

# Developments on a 22GHz Microwave Radiometer and Reprocessing of 13-Year Time Series for Water Vapour Studies

Alistair Bell[a,b], Eric Sauvageat[a,b,c], Gunter Stober[a,b], Klemens Hocke[a,b], and Axel Murk[a,b]

[a]Institute of Applied Physics, University of Bern, Bern, Switzerland
[b]Oeschger Centre for Climate Change Research, University of Bern, Bern, Switzerland
[c]Federal Office for Meteorology and Climatology MeteoSwiss (Since September 2023).

**Correspondence:** Alistair Bell (alistair.bell@unibe.ch)

**Abstract.** Long-term observations of water vapour in the middle atmosphere are important for climate studies and predictions, chemical and dynamical process studies, as well as modelling certain weather events with implications for surface conditions. Measurements from an instrument making middle atmosphere water vapour observations near Bern, Switzerland- named MIAWARA- have been completely reprocessed since 2010. This has comprised of a new calibration which has been integrated into the framework for the calibration of other University of Bern radiometers, and a new retrieval algorithm. The installation of a new spectrometer on the instrument has also allowed the comparison and correction of past observations. We present these corrected measurements and their subsequent analysis against data from Aura MLS. The comparison shows that the corrected spectra yield more consistent values of water vapour mixing ratio between MIAWARA and Aura MLS, with a lower standard deviation of differences at all heights, and a reduced bias between the two instruments at pressure (height) levels below (above) 0.3 hPa.

## 1 Introduction

Changes in middle atmosphere water vapour can result in important radiative forcings relevant to changes in global surface temperature (Solomon et al., 2010). Fluctuations will also affect temperatures in the middle atmosphere (Maycock et al., 2011), and can directly or indirectly affect the concentrations of other gases (Brasseur and Solomon, 2005; Thölix et al., 2018). The mixing ratio of water vapour and horizontal and vertical gradients of water vapour are important for the circulations in the middle atmosphere, such as the Brewer-Dobson circulation, and changes in water vapour are commonly used as a tracer for such circulations (Straub et al., 2012; Hocke et al., 2018; Shi et al., 2023).

Whilst there are many methods of measuring water vapour in the troposphere and lower stratosphere (Brunamonti et al., 2018; Hicks-Jalali et al., 2020), methods of measuring water vapour in the upper stratosphere and mesosphere are much rarer, owing to both the distance away from the surface, the harsh physical conditions present in stratosphere and mesosphere, and the relatively small quantities present relative to those typically measured in the troposphere. Despite semi-routine measurements in the lower-stratosphere being possible with in-situ balloon-borne devices (Hurst et al., 2011; Graf et al., 2021; Brunamonti et al., 2023), most measurements rely on remote sensing technologies (Sica and Haefele, 2016). A common method for both





ground-based and satellite-borne instruments is microwave radiometry (Haefele et al., 2008; Straub et al., 2012; Schranz et al., 2019).

Whilst water vapour in the troposphere has been relatively well sampled, for the purpose of process studies, weather forecasting and climate monitoring, measurements of water vapour in the middle atmosphere are much more sparse. A key source of information about global water vapour in the middle atmosphere comes from the Microwave limb sounder on board the Aura satellite (Waters et al., 2006), which was launched in 2004. However, as with many instruments, aging of components has led to measurement drifts (Livesey et al., 2021), and it can be difficult to distinguish drifts as a result of instrumental artifacts from drifts borne out of atmospheric changes.

For instruments on board satellites, diagnosing and fixing instrumental problems can be even more difficult due to the remote nature of the instrument. It is hence important to maintain a network of instruments, used with a collaborative approach to ensure that data are reliable and to reduce the impact of individual instrument errors.

Although routine global measurements of water vapour in the middle atmosphere have been made with the microwave limb sounder on the Aura satellite since 2004, these observations face several drawbacks when compared to ground-based upward-looking radiometers. The Aura-MLS has a revisit time of twice per day at fixed local time, meaning that observations can be sparse, and there is no possibility of capturing diurnal variations. As satellite instruments cannot be serviced, instrumental aging which can cause drifts in the measurements can be difficult to separate from atmospheric trends, thus it can be difficult to separate instrumental artifacts from measured values. The lifetime of a satellite is also finite; the satellite may fail due to aged electrics, fuel depletion, or a range of other factors. The MLS instrument will end its lifetime in 2026, with the water vapour measurements severely restricted from Spring 2024 (NASA Jet Propulsion Laboratory, 2024). Currently, there are no scheduled launches for a replacement instrument, which means that the most valuable resource for monitoring water vapour will be the network of ground-based radiometers.

## 2 MIAWARA

The MIddle Atmosphere WAter vapour RAdiometer (MIAWARA) is a 22 GHz radiometer that has been operated by the Institute of Applied Physics (IAP) at the University of Bern since April 2002 (Deuber et al., 2004). It was installed at the Zimmerwald Observatory near Bern in 2006 and has since provided long-term observations of this essential climate variable, in the stratosphere and lower mesosphere.

The MIAWARA records the downwelling spectral radiance around the 22 GHz $H_2O$ absorption line. This absorption line is a narrow range of frequencies across which radiation is increasingly absorbed by the air layer which it is passing through; and, through Kirchoffs law, increasingly emitted. As the intensity of this absorption line increases with the water vapour mixing ratio, surface radiance can be used to retrieve the water vapor mixing ratio. Due to the effect of pressure broadening (Liebe, 1985), which increases the range of frequencies over which the absorption line contributes significantly to surface spectral radiance with the (atmospheric) pressure, recorded spectral radiances can be inverted to retrieve vertical profiles of water vapour.





The instrument has a corrugated feed horn antenna which is directed towards a plane mirror. The mirror rotates, allowing a range of elevation angles and calibration targets to be observed, such as the reference target, hot load, and liquid nitrogen target. As reflections between the receiver and external targets (notably the reference target) will cause unwanted standing

wave errors, the mirror is mounted on a linear translation stage which alternates the distance from the horn antenna between the observations by a quarter wavelength. This changes the phase of the standing waves, and by averaging the data of the two positions the standing wave errors are reduced (Gustincic, 1977)

## 2.1 Calibration

As the 22 GHz line has a fairly small magnitude (of the order 0.1K), measurement noise and spectral non-linearities can severely

affect the retrieved water vapour mixing ratio, the measurement response, and the effective height range of the retrieval. In addition, the brightness temperatures observed in this frequency range, which are typically between 30K to 70K at the zenith, are considerably lower than those of other frequently monitored microwave absorption lines, like the 183 GHz water vapour line, or those associated with ozone and oxygen.

In order to optimise the noise and linearity of the calibrated spectra, most middle atmosphere water vapour radiometers use

a balancing calibration scheme (Forkman et al., 2003). This involves using a reference view which produces a signal of around the same intensity as the sky observation. For this radiometer, the sky view is at an angle that is calculated to optimise the middle atmospheric emission, whilst not being so low as to have excessive attenuation from the troposphere. This is an elevation angle that is typically between $15°$ to $30°$, depending on the optical depth of the troposphere, which tends to be seasonally dependent. The reference observation is then the sky at zenith view partially blocked by a strip of microwave absorber. The combination

of the sky at zenith, which is cold compared to the observation at a lower elevation angle, and the absorber, which is at ambient temperature, and much warmer than the observed sky radiances, means that a signal of around the same magnitude as the line observation can be made. The angle of the view is adjusted by up to 14 degrees, allowing more or less of a contribution from the hot absorber to be contained in the antenna view, as less of the absorber is seen by the antenna at lower elevation angles.

Raw measurements are made in "counts", a quantisation of the analog signal following conversion from the ADC (analog

to digital converter). In order to calibrate these measurements to the more useful unit of brightness temperature, hot and cold observations, with a known physical temperature are required. For this, a microwave absorber at ambient temperature is used for the hot target, and the sky at a high elevation angle is used for the cold target. The brightness temperature of the sky at this elevation angle is calculated by taking several further measurements at different elevation angles, and performing a tipping curve calibration (Han and Westwater, 2000). This tipping curve measurement is currently performed once every 15 minutes

and contains seven different elevation angles, though the number of angles, the period between successive tipping curves, and integration time at each observation angle have been changed slightly over the years.

The difference between the line and reference counts is calibrated with the differences in hot and cold temperature and hot and cold counts according to equation 1. This brightness temperature difference is then corrected for tropospheric attenuation by a factor calculated from the tropospheric optical depth calculated previously.





$$\Delta Tb = \frac{(U_{\text{Line}} - U_{\text{Ref}})}{(U_{\text{Hot}} - U_{\text{Cold}})} \cdot (T_{\text{Hot}} - T_{\text{Cold}}) \tag{1}$$


From Ingold et al. (1998), the downwelling brightness temperature at the tropopause can be estimated by modelling the atmosphere as a two-layer system. Although in the calibration, emission from the troposphere is not corrected for- as this is easily corrected for in the retrieval step- the attenuation of the signal coming from middle atmospheric emission by the troposphere is corrected for. This step allows the retrieval to assume an instrument altitude above the tropopause, meaning that the troposphere does not need to be taken into account in the radiative transfer model.


A correction factor $a$, which corrects for the tropospheric absorption based on the previously calculated optical depth, can be applied. The formula also corrects for the relative air masses of the troposphere and middle atmosphere viewed by the reference observation ($AM_{\_\text{ref\_trop}}$ and $AM_{\_\text{ref\_ma}}$) and the line observation ($AM_{\_\text{line\_trop}}$ and $AM_{\_\text{line\_ma}}$). The product of the correction factor and the brightness temperature difference then gives a brightness temperature difference equivalent to a zenith angle view (i.e. with an air mass factor of one).


$$a = \frac{1}{AM_{\text{line\_ma}} \cdot \exp(-\tau \cdot AM_{\text{line\_trop}}) - A \cdot AM_{\text{ref\_ma}} \cdot \exp(-\tau \cdot AM_{\text{ref\_trop}})} \tag{2}$$

$$Tb_{\text{Bal}} = a \cdot \Delta Tb \tag{3}$$

Among other weather conditions that can lead to increased noise in the calibrated spectra, precipitation has the greatest impact. When the presence of precipitation is sensed by a rain sensor, the rain hood on the instrument closes. Although the instrument still runs, there is now a rain-covered radom between the sky and the antenna, adding effects of attenuation and reflection to the measurements. As readings containing rainy data tend to increase the noise of a calibrated spectra for a given day, all readings taken when the rain hood is closed are excluded from the calibrations.


For the reprocessed data set, it was found that an optimal integration time of one day would be used for retrievals. This is a compromise between having a good enough temporal resolution to resolve changes with time, and having a low enough noise in the calibrated spectra, that a retrieved profile with a large enough height range could be made.


## 2.2 Retrieval

The retrieval of the water vapour profile from the calibrated spectra is handled by the radiative transfer software ARTS (Buehler et al., 2018), which is integrated into the Matlab software Qpack (Buehler et al., 2018). The Levenburg-Marquart algorithm calculates updates of the water vapor state $x$ upon each new iteration within the algorithm.


The retrieval of water vapour in the middle atmosphere relies on a high spectral resolution, and as such, across a $1\,\text{GHz}$ bandwidth, brightness temperatures at several thousand frequencies are calibrated (see Table 1). In order to compute the forward model with such a large number of frequencies, which is calculated at each iteration of the optimal estimation algorithm, a



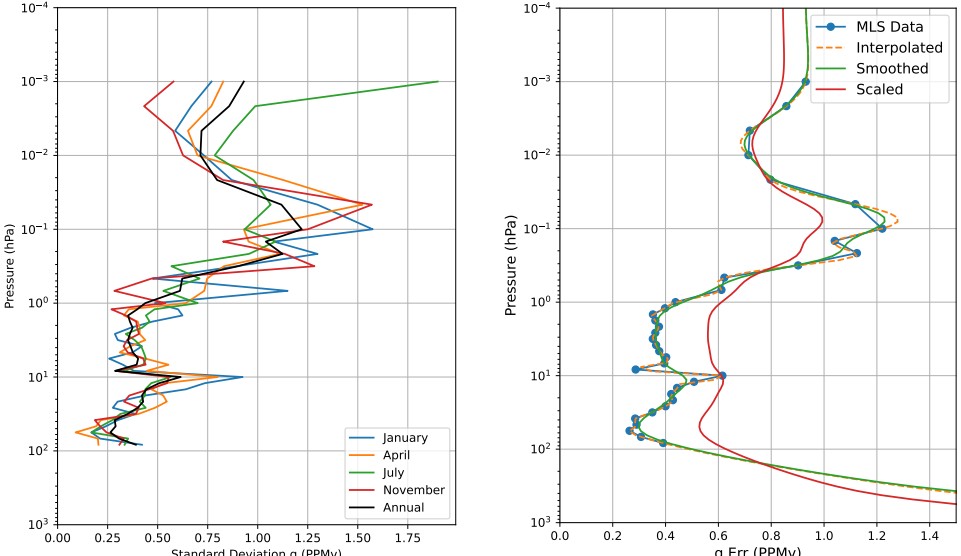

**Figure 1.** Standard deviation across pressure levels found from MLS data, and the yearly, smoothed data used to generate the error covariance matrix in the retrieval algorithm.

non-trivial amount of computational power is required. In order to speed up retrieval time, the forward model is run at the frequency resolution of the instrument at the central frequency, and then the spacing between the channels is increased with distance from the 22.235 GHz absorption line frequency so that only 1200 frequencies are simulated altogether.


For the optimal estimation approach (Rodgers, 2000), a first guess or *a priori* is needed. This is what can be considered the best estimate of the retrieval state *x*, which in this case is a vertical profile of the water vapour mixing ratio. In this case, two sources of information were chosen for the *a priori* profile: the ECMWF analysis of 6 years (2010 - 2015) was used. The average zonal monthly mixing ratios were computed from this.


As the height range of this forecast only extends as far as 0.024 hPa, the MLS data set (see section 3.4) is also used for heights above 0.5 hPa. The two profiles are used between 0.5 hPa and 0.024 hPa, over which pressures a cubic smoothing spline function is applied (matlab function `csaps` based on De Boor and De Boor (1978)). To compute values for a specific date, an interpolation of the average monthly zonal H2O from the two nearest months to that date (e.g., averages from April and May for a date of April 30th) is performed, considering their distances from the target date.


Another important source of error comes from the estimation of the *a priori* error covariance matrix. For the retrieval framework here, a full matrix is not prescribed, however, a profile of estimated standard deviations along with a correlation length of 0.25. An estimation of the *a priori* state for each month was performed by calculating the standard deviation of water vapour above Bern measured on the first day of each month between 2010 and 2018. As shown on figure 1, despite some differences between months at different levels, there was no significant seasonal differences. Because of this, a static profile


of standard deviations is used to generate the *a priori* error covariance matrix, generated from the average standard deviations




from all 12 months. In order to generate a more continuous profile from the calculated standard deviations, additional points are added to the vertical axis, and interpolation is performed using the SciPy `interp1` function with a cubic smoothing function (Contributors, 2023). This is further smoothed by convolving the profile with a Gaussian kernel to reduce the remaining oscillations. Before being used in the retrieval algorithm, the profile is further increased by 0.2 PPMv at each level to account

for resolution/grid spacing errors and integration time differences which are not accounted for in the yearly variability. An extra requirement is made of the *a priori* standard deviations that they are between $20\,\%$ and $80\,\%$ of the a priori profile at any given level. Should the errors fall outside of this range, they are increased or decreased until they are in agreement with it.

The correction for the tropospheric attenuation of the middle atmospheric signal allows the retrieval surface height to be set at $15\,\mathrm{km}$. A bandwidth of only 70MHz is used for the retrieval, as bandwidths larger than this, whilst improving the effective

range of the retrievals, were seen to cause additional problems in managing the spectral baselines (such as in Nedoluha et al. (2013); Verdes et al. (2002)), and could often lead to degraded retrievals. To account for any mismatch in the frequency axis, a frequency shift is retrieved. The spectral baseline, which is often seen in the calibrated spectra, is a major problem which limits both the height range and the accuracy of retrievals. This is a known problem, and can come from numerous sources, such as the standing waves as discussed in section 2. For several reasons, the shift in mirror position is not able to completely remove

such artifacts. This can be due to the change in path length meaning that different views of the reference bar are seen with the different positions, and several other reasons which are discussed in great detail in Deuber and Kämpfer (2004).

In addition, although the tropospheric attenuation affecting the signal from the middle atmosphere is accounted for in the calibration step, the emission from the troposphere is not corrected for. This means that the balanced brightness temperature contains a baseline contribution from the tropospheric emission. In order to remove this baseline, a second-degree polynomial

fit is retrieved inside the optimal estimation algorithm.

The spectral baseline which affects measurements may be irregular, non symmetric, and change over time, all of which make it difficult to systematically remove. Although a common approach to this problem is to try to fit sinusoidal functions to the brightness temperature spectra, such as Sauvageat et al. (2022), an issue with this approach in this instance was "over fitting". This can happen when there are features in the spectra that could be fitted by a sinusoidal function over a given frequency

range, but not over the whole spectra. This can then have the effect of creating features in the spectra, which in the optimal estimation can create something non-physical in the retrieved profile.

## 2.3   Updates

Several hardware upgrades have been made to the instrument since its first observations in 2001. The most notable of these has been to the spectrometer. The spectrometer is a key component of the receiver, which converts the time-domain electrical

signal into a frequency domain signal. The working bandwidth, frequency resolution, and linearity of the spectrometer have large implications for the quality of the water-vapour profile that can be retrieved.

As developments in spectrometer hardware have been made over the lifetime of MIAWARA, the spectrometers have been updated. As the instrument has been designed to accommodate up to three spectrometers, this allows an intercomparison of the calibrated spectra and brightness temperatures to be made between spectrometers.



**Table 1.** Specifications of the main spectrometers used onboard the MIAWARA instrument.

| | Chirp Transform | AOS | AC 240 | USRP |
|---|---|---|---|---|
| **Manufacturer** | Paul Hartogh (MPS) and University of Bern | Observatoire de Meudon | Aquiris | Ettus Research |
| **Bandwidth** | 40 MHz | 1.1 GHz | 1 GHz | 2 x 200 MHz |
| **No. Channels** | 4200 | 1725 | 16384 | 2 x 16384 |
| **Channel Spacing** | 9.5 kHz | 610 kHz | 61 kHz | 12.2 kHz |
| **Frequency Resolution** | 14.07 kHz | 1.1 MHz | 61 kHz | 12.2 kHz |
| **Reference** | Hartogh (1997) | Jost et al. (1996) | Benz et al. (2005) | Hagen et al. (2020) |

Originally, two spectrometers were installed on the instrument (Deuber et al., 2004), an acousto-optical spectrometer for broadband analysis, and a chirp-transform spectrometer with 4200 channels for precise analysis of the line center. Technological advances from FFT spectrometers have been shown to provide significant improvements over AOS and filterbank spectrometers in terms of resolution, bandwidth, and system stability, compared to AOS and filterbank spectrometers (Muller et al., 2009). For this reason, a broadband FFT spectrometer, the Aquiris AC-240, was also installed on the instrument in 2007

and is due to record measurements until at least the end of 2024. However, the Aquiris AC-240 has recently been shown to have issues with non-linearity (Sauvageat et al., 2021), which is examined in this article in section 3. In November 2022, a new FFT spectrometer was installed, based on the software-defined radio (SDR) USRP X310 manufactured by Ettus Research (Hagen et al., 2020). It has two input receivers, which can be tuned over a frequency range from $10\,\mathrm{MHz}$ to $6\,\mathrm{GHz}$. Both channels have a usable bandwidth of slightly under $200\,\mathrm{MHz}$, giving a $12.2\,\mathrm{kHz}$ channel spacing. The spectrometer demon-

strates reduced bias and improved linearity compared to the AC240. This improvement stems from the 14-bit analog-to-digital converter (ADC), a notable step up from the AC240's 8-bit ADC, as well as advancements in digital signal processing that reduce numerical errors. The main features of each spectrometer is summarised in table 1.

## 2.4   Reprocessing

The raw spectra obtained from the instrument have been reprocessed from October 2010 until October 2023. The new cali-

bration routine has been integrated into a unified calibration framework for radiometers managed by the University of Bern (Sauvageat, 2021). Importantly, additional quality checks have been added to the calibration cycle (e.g. noise in spectra too high, median brightness temperature outside of 3 standard deviations of other median brightness temperatures in integration time, mean balanced brightness temperature being less than -5K or more than 5K). Where these quality checks are not passed, the spectra are not included in the integrated measurement.



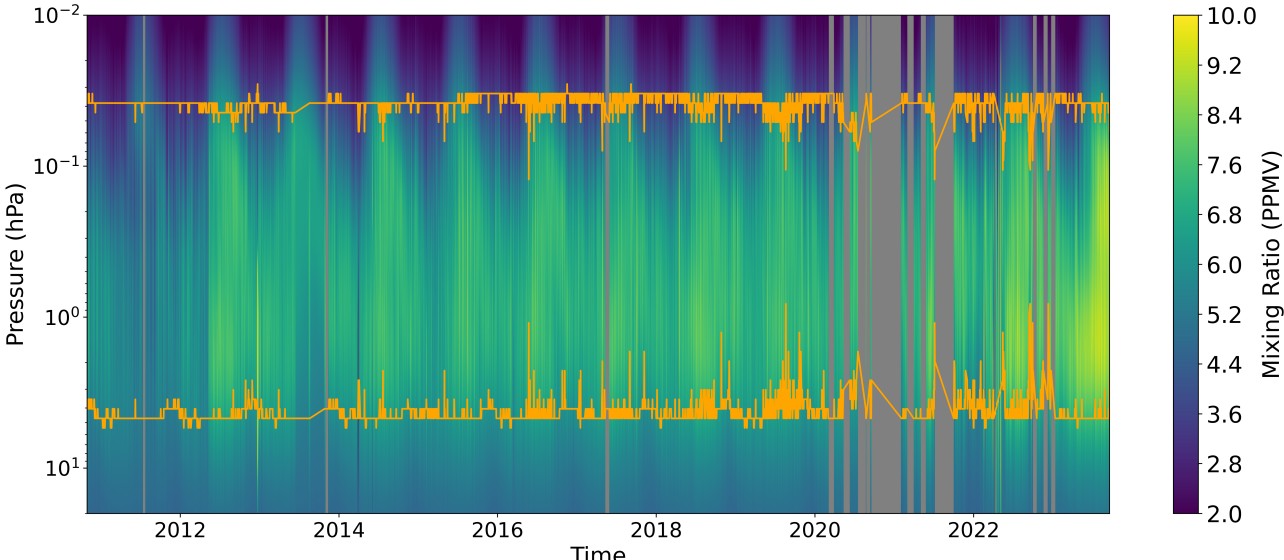

**Figure 2.** Reprocessed MIAWARA time series from October 2010 until October 2023, all made with the AC-240 spectrometer (right).

Although in the past, variable integration times have been used based on the spectral noise (Lainer et al., 2018), for the reprocessed data it has been considered more useful to have a dataset with a consistent temporal resolution. For this reason, the level 1b data (calibrated, integrated data with quality checks) has been processed and integrated to a temporal resolution of one spectra per day. It was found that with integration times of less than one day, on days with less than optimal observing conditions, that the spectral noise could noticeably affect the retrieval range. Conversely, integration times of more than one day did not lead to a significant improvement in retrieval quality, whist limiting the usefulness of such a dataset.

## 3 Comparative Performance of AC-240 and USRP Spectrometers

The USRP spectrometer was installed on the instrument in November 2022. At the same time, a number of maintenance tasks, such as the replacement of the reference bar absorber, were also performed. Although the spectrometer provided measurements for the first few days after installation, in November there was a software problem that led to the spectrometer not recording observations again until the start of 2023.

### 3.1 Calibrated Spectra

In order to verify the new spectrometer, the calibrated spectra from the two were compared for a liquid nitrogen (LN2) calibration made in March 2022. During the LN2 calibration, observations were taken of the hot load, LN2 target, cold sky, reference



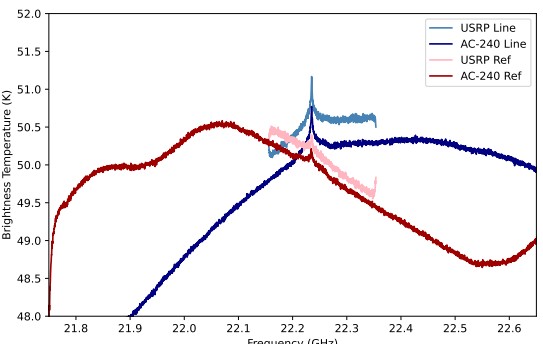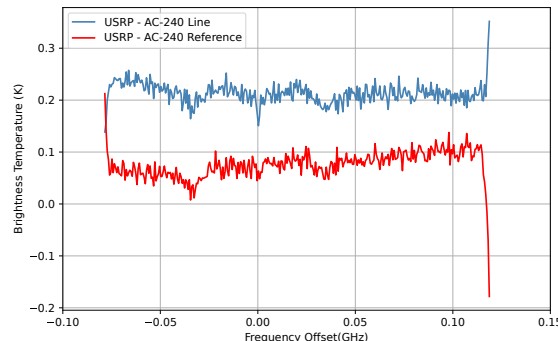

**Figure 3.** Measurements of the line and reference observation views calibrated with an LN2 calibration taken as recorded on the AC-240 and USRP spectrometers (left) and the difference between the two spectrometers for two observation views.

and line views, which allowed a calibration of the line and reference views which is independent of the tipping curve calibration. Figure 3 shows the calibrated line and reference observations measured on the two spectrometers, where line observations are shown in blue and reference in red. One can see from this image that the reference observation is not exactly that which one would expect from the theory presented. Instead, there seems to be a sinusoidal oscillation in the spectra between $21.7\,\mathrm{GHz}$ and $22.1\,\mathrm{GHz}$, after which there is a linear decrease in brightness temperature up to around $22.5\,\mathrm{GHz}$. This is likely due to the frequency-dependency of the antenna pattern, causing a larger proportion of the hot absorber to be seen at some frequency ranges compared to others.

It may be seen that despite the vertical shift of the spectra by around $0.1\,\mathrm{K}$ for the reference observation, and $0.2\,\mathrm{K}$ for the line observation, that the spectra agree in shape fairly well. The differences between the measurements recorded on the USRP and AC-240 were then computed, binning the USRP measurements by 50 frequency vectors and the AC-240 by 10 frequency vectors. When the differences between the spectra are computed for each frequency, some features become more evident. For example, there is a feature that is clear on both the line and reference measurements at a frequency offset of $-25\,\mathrm{MHz}$ and $25\,\mathrm{MHz}$, where there appears to be a slight oscillation. This is caused by the non-linearity of the AC-240 spectrometer, and is visible if one looks closely at the line and reference spectra. Though this feature cannot be considered to be ideal, as it appears in the reference and line observation, and the difference between these spectra is used for the retrieval of water vapour, the feature is cancelled out, and the brightness temperatures in this frequency range may be safely used in the retrieval algorithm.

It may also be seen in this figure that at the line centre, there is a decrease in the bias. This is most likely caused by the central two channels of the AC-240 that do not give accurate readings. In the updated retrieval framework, these two central frequencies are not used. Though the non-linearities present in the AC-240 spectrometer are evident from this comparison, a further known bias, due to spectral leakage (Sauvageat et al., 2021), is not evident. This was potentially due to the relatively large noise levels in these calibrations due to the short integration times of several hours, compared to the operational integration time of one day.





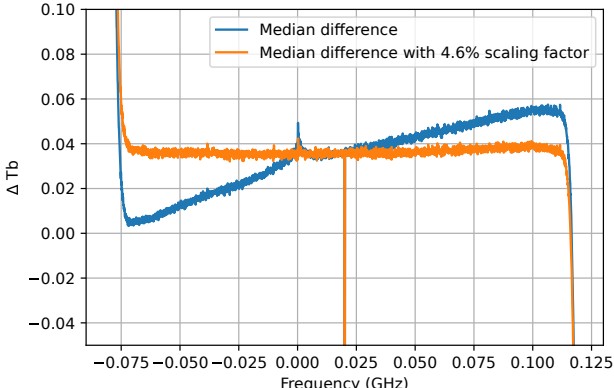

**Figure 4.** The differences in balanced brightness temperature between measured by the USRP spectrometer and the AC-240 (USRP - AC-240), and the USRP and the AC-240 when a scaling factor of $4.6\,\%$ is applied to the spectra.

In order to compare the two spectrometers with a minimised noise level, one year of calibrated balanced brightness temperature measurements were used, between the installation of the spectrometer in November 2022 and October 2023. Due to several software issues, the USRP did not make readings for the whole time period and thus a data set of 265 days on which measurements had been made on both spectrometers were used. Readings for which one of the spectrometers contained readings outside of three standard deviations for more than $5\,\%$ of the spectra were discarded. With the rest of the data, the median brightness temperature at each frequency on each spectrometer was computed. The differences between the two spectrometers was then found. This result is shown in figure 4. One can see in this figure that the USRP records throughout the frequency range larger values than the AC-240. It can also be seen that the features of the calibrated spectra are clear on this plot of the differences between spectrometers, indicating that the bias is proportional to the brightness temperature itself. According to the methodology presented in Sauvageat et al. (2021), the effects of spectral leakage may be corrected for by using equation 4:

$$T_{B,corr} = \frac{1}{(1-\alpha)}(T_B - \overline{T}_B - \Delta T_{B,c}) \tag{4}$$

Where $T_{B,corr}$ is the brightness temperature corrected for the effects of spectral leakage, $\alpha$ is the scaling factor, $\overline{T}_B$ is the mean brightness temperature from a single measurement across all frequencies, and $\Delta T_{B,c}$ is a non-linearity correction. The non-linearity correction is an additional correction based on the absolute brightness temperature, and was found to amount to a maximum of around $0.2\,\mathrm{K}$ for a brightness temperature increase of $100\,\mathrm{K}$. As this is a relatively minor amount for the range of calibrated brightness temperatures used in the retrievals, which tend to have a range on the order of $1\,\mathrm{K}$, the non-linearity term here was not considered. When the AC-240 was scaled by $4.6\,\%$, both the slope and the line feature are removed. This indicates that the problem of spectral leakage is present in the AC-240 and that it could be corrected for by a $4.6\,\%$ scaling factor.



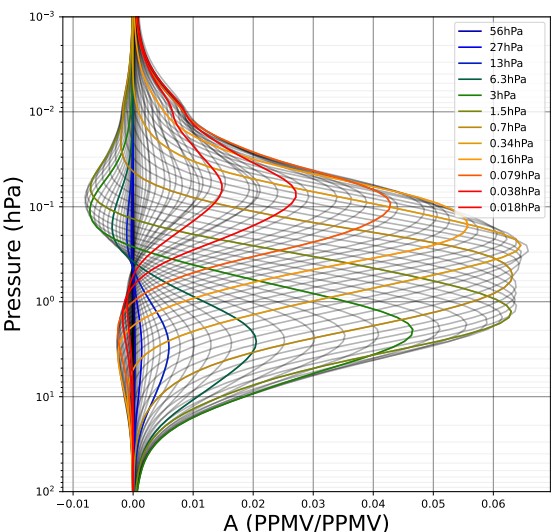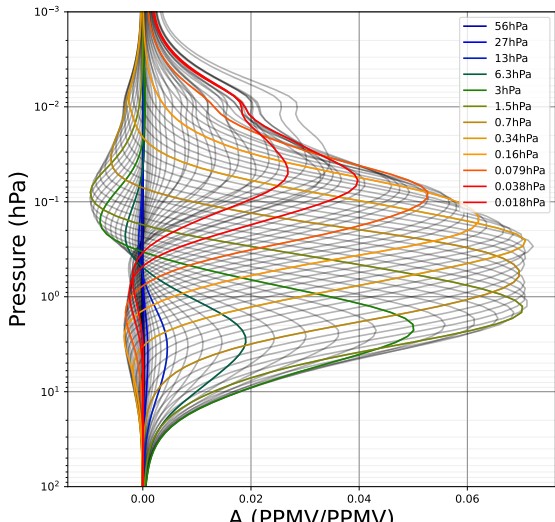

**Figure 5.** Mean averaging kernel for retrievals made with the AC-240 (left) and USRP (right) spectrometers between November 2022 and October 2023.

## 3.2 Retrieval Range and Resolution

One key advantage of the USRP over the AC-240 in making retrievals of middle atmosphere water vapour is the spectral resolution, which is five times higher than the AC-240 (see table 1). A higher spectral resolution could in theory allow a larger retrieval range, as the centre of the absorption line, from which information about the water vapour mixing ratio higher up in the atmosphere comes, is better resolved. One way of judging the information content by height can be gained by looking at the averaging kernel from the retrieval. The averaging kernel represents the sensitivity of the retrieved state to changes in the true state. An ideal averaging kernel would therefore be an identity matrix. However, as information is spread across many retrieval levels, this is not the case.

From the plots of the two averaging kernels, shown in figure 5, one can see that similar values are found at the maximum sensitivity of averaging kernels up to around $0.04\,\text{hPa}$. Above the height corresponding to this pressure level, retrievals from the USRP spectrometer have a higher sensitivity than the AC-240, although the sensitivity is around half that found between $1\,\text{hPa}$ and $0.1\,\text{hPa}$. The averaging kernels at pressures below $0.02\,\text{hPa}$ also seem to have a bimodal distribution and have a certain skewness.

The value of the averaging kernel at maximum sensitivity gives information about the validity of the retrieval level at which this occurs, and the full width at half maximum (FWHM) for each retrieval level can also be of interest. If the maximum of a column is located along the diagonal, it indicates that the retrieved level has the highest sensitivity to its corresponding true level. This is a good sign as it means the retrieval is most sensitive to the actual level that it is supposed to represent. The





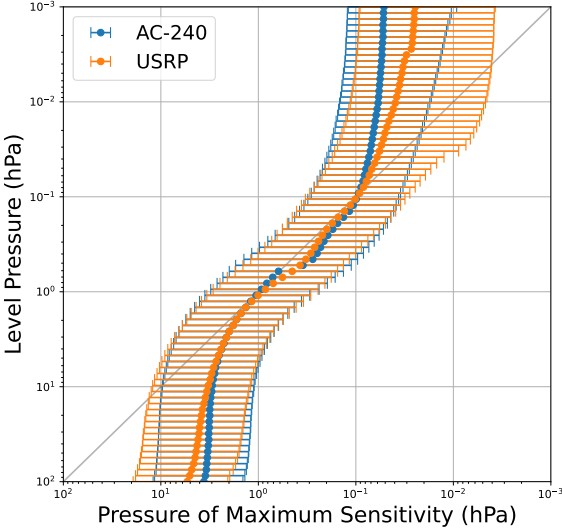

**Figure 6.** Pressure of maximum sensitivity of retrieval to changes in the true state, with the error bars representing the pressure at which the curve is equal to half the maximum value.

wider the FWHM, the wider the information is spread among the retrieved states, hence the lower the vertical resolution of the retrieval.

Figure 6 shows that between $2\,\mathrm{hPa}$ and $0.08\,\mathrm{hPa}$, the averaging kernel for both the AC-240 and the USRP is centered very close to the retrieval pressure level, as desired. At lower pressures, retrievals from both spectrometers have the maximum sensitivity at pressure levels below the retrieval level. However, the retrieval level from the USRP measurements stays within the FWHM up to $0.004\,\mathrm{hPa}$ whereas for the AC-240, the FWHM stays within the retrieval level pressure only up until $0.020\,\mathrm{hPa}$. Though it must be highlighted that, as shown in figure 5, the sensitivity at these higher levels is approximately a third of that at $1\,\mathrm{hPa}$. The FWHM between $1\,\mathrm{hPa}$ and $0.1\,\mathrm{hPa}$ is slightly improved with the USRP, which is demonstrated with the smaller error bars. From these findings, it would be expected that retrievals with the new spectrometer be more representative than retrievals made with the AC-240 spectrometer for water vapour deviations from the a priori above $0.1\,\mathrm{hPa}$.

### 3.3 Retrieved Water Vapour

The differences in the averaging kernel of the retrieval and differing spectrometer properties will impact the values of retrieved water vapour. As the retrieval is handled by virtually the same algorithm for measurements made by both spectrometers, save for some adaptions to the instrumental errors to take into account different frequency resolutions, retrievals from observations made with both spectrometers in parallel can be directly compared, and differences directly attributed to the hardware difference.





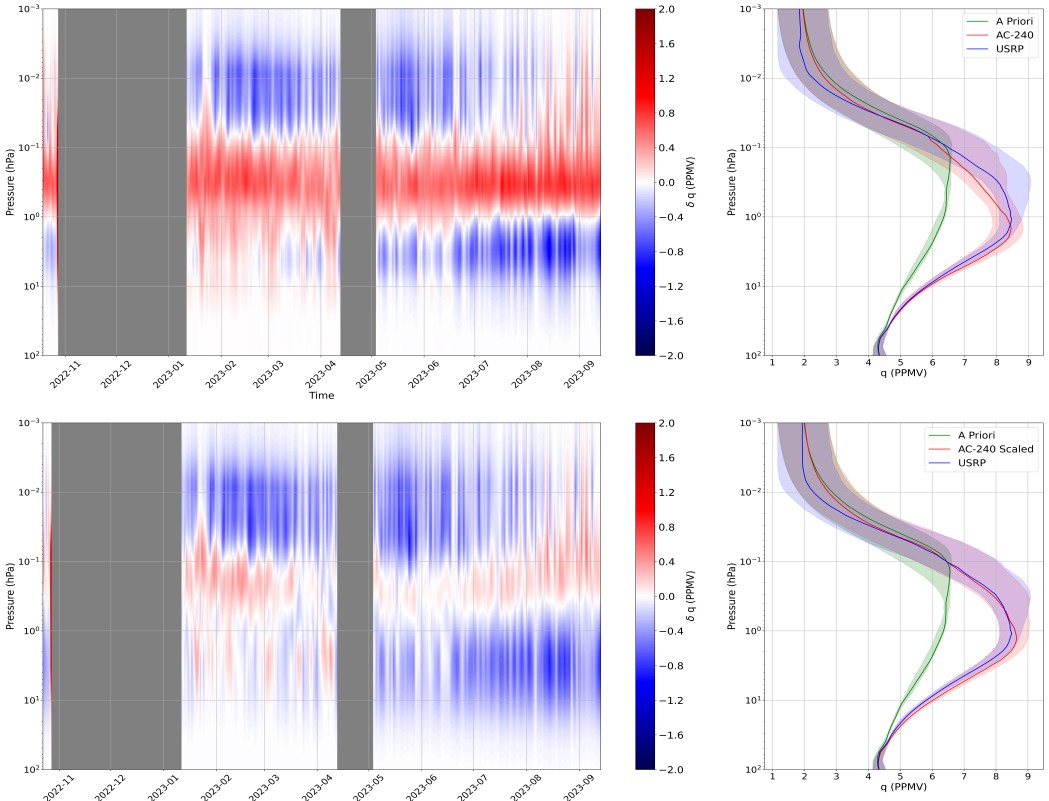

**Figure 7.** The difference between profiles of water vapour measured with the USRP and the AC-240 spectrometer (USRP - AC-240), between the installation date in November 2022 and October 2023 (top left); the median with 25th and 75th percentiles (in translucent) of *a priori*, retrieved values from AC-240 measurements, and retrieved values from USRP measurements (top right). The same plots are shown again on the bottom row, but for AC-240 measurements that are corrected with the scaling factor.

As is show in figure 7, there is a bias of below $1\,\mathrm{PPMv}$ in the profiles retrieved with the AC-240 relative to those retrieved with the USRP between $1\,\mathrm{hPa}$ and $0.1\,\mathrm{hPa}$ that persists between the start and end of the period. The AC-240 also contains a negative bias in spring at pressures around $0.01\,\mathrm{hPa}$, whilst in the summer this bias fades and is replaced by a negative bias at pressures around $4\,\mathrm{hPa}$.

Despite the fact that the differences between the two spectrometers appears to be significant, by comparing the distribution of retrieved values to the *a priori* climatology, it may be seen that the differences between retrieved profiles are relatively small compared to the differences between retrieval and climatology. This is influenced by the fact that 2022 and 2023 have been marked by significantly higher levels of water vapour in the stratosphere and mesosphere compared to average (Nedoluha et al., 2023). In fact the median values agree very well up to $1\,\mathrm{hPa}$, above which they diverge slightly to have a maximum difference in median mixing ration at $0.34\,\mathrm{hPa}$ of $0.62\,\mathrm{PPMv}$.



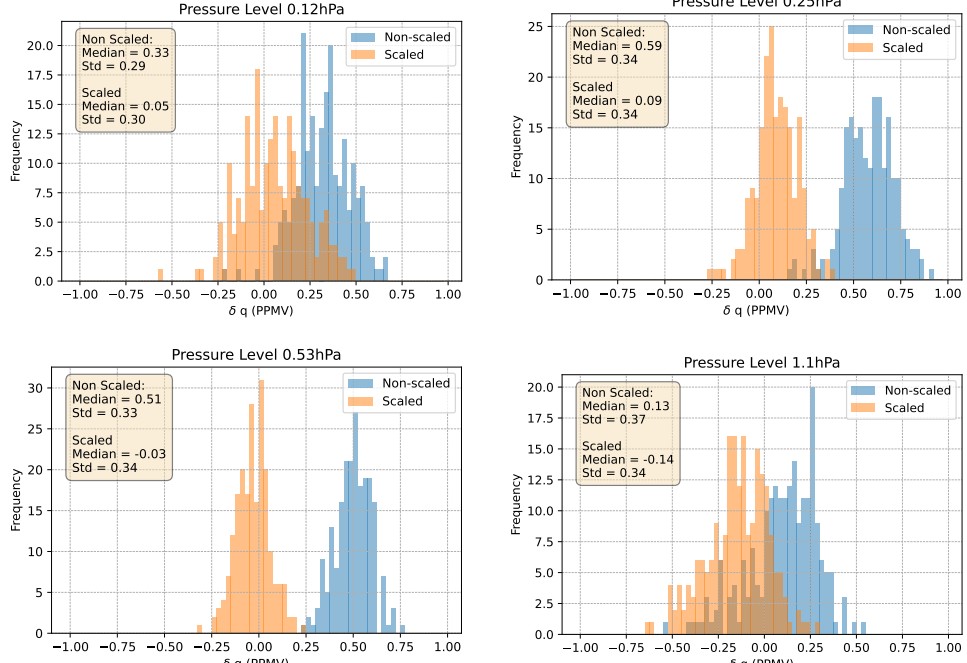

**Figure 8.** Histograms of the difference in the retrieved water vapour with USRP minus AC-240, where AC-240 retrievals with the corrected measurements, made with a $4.6\,\%$ scaling factor, are compared to the uncorrected measurements. These are shown at pressure levels indicated in the title of each subplot.

Comparing the differences in retrieved water vapour from the non-scaled AC-240 measurements and the scaled AC-240 measurements can show whether the retrieved values become more consistent with those from the USRP and not simply that that the median retrieval is more similar to that of the USRP. In figure 8, the distribution of the differences between the unscaled AC-240 retrievals and the USRP retrievals, and the scaled AC-240 retrievals and the USRP retrievals, all at four pressure levels between 0.1 hPa and 1 hPa.

### 3.4 Comparison to MLS

The Microwave Limb Sounder (MLS) on board the Aura satellite is a microwave radiometer containing five receivers with a total of 1308 frequency channels, which allow the retrieval of at least 15 atmospheric constituents, plus temperature profiles, at height ranges varying by species from up to $80\,\mathrm{km}$ down to $5\,\mathrm{km}$ (Waters et al., 2006). Due to the frequency of measurements and the possibility of co-located measurements, MLS is often used as a reference to compare other middle atmosphere measurements to. Despite this, MLS observations have been subject to an instrumental drift, which though corrected for in the version five (V5) of the measurement dataset, still has a drift of when compared to frost point hygrometers. They do, however now exhibit no drift when compared to the atmospheric Chemistry Experiment Fourier Transform Spectrometer (ACE-FTS) (Livesey et al., 2021).



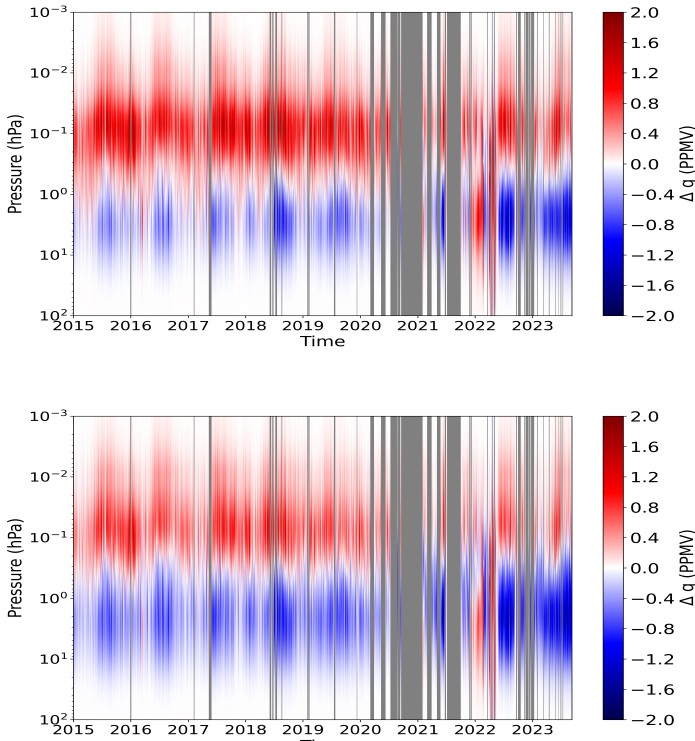

**Figure 9.** Time series of the difference between MLS water vapour and MIAWARA measurements made with the uncorrected spectra (top) and the corrected spectra (bottom).

In order to verify the reprocessed retrieval, and whether the re-scaled calibrated spectra agree better with other measurements of middle atmosphere water vapour, MLS was thus used for comparison. From Livesey et al. (2022), due to the high vertical resolution of MLS measurements compared to other remotely sensed observations, it is common to compare measurements without reference to the MLS averaging kernel. This was the approach that was taken in this work, and to compare MIAWARA meaurements to MLS, the MLS profile was convolved with the MIAWARA averaging kernel. A eight-year dataset from January 2015 until October 2023 was created, for which period both MLS observations and MIAWARA observations were available.

Figure 9 shows the differences between the MLS and MIAWARA measurements from the period between 2015 and 2023. There appears to be an annual pattern to the differences in retrievals, with biases becoming greater in the summer period compared to the winter. This pattern is the same for both the uncorrected and corrected AC-240 spectra. It can also be seen that the negative bias, between $10\,\mathrm{hPa}$ and $1\,\mathrm{hPa}$ from 2022 until the end of the period, becomes greater, coinciding with above average water vapour mixing ratios seen throughout this period. When the MLS differences with the uncorrected spectra and the corrected spectra are compared, it can be seen that the positive bias between $0.1\,\mathrm{hPa}$ and $0.2\,\mathrm{hPa}$ is less intense with the corrected spectra, whilst the negative bias is slightly increased.



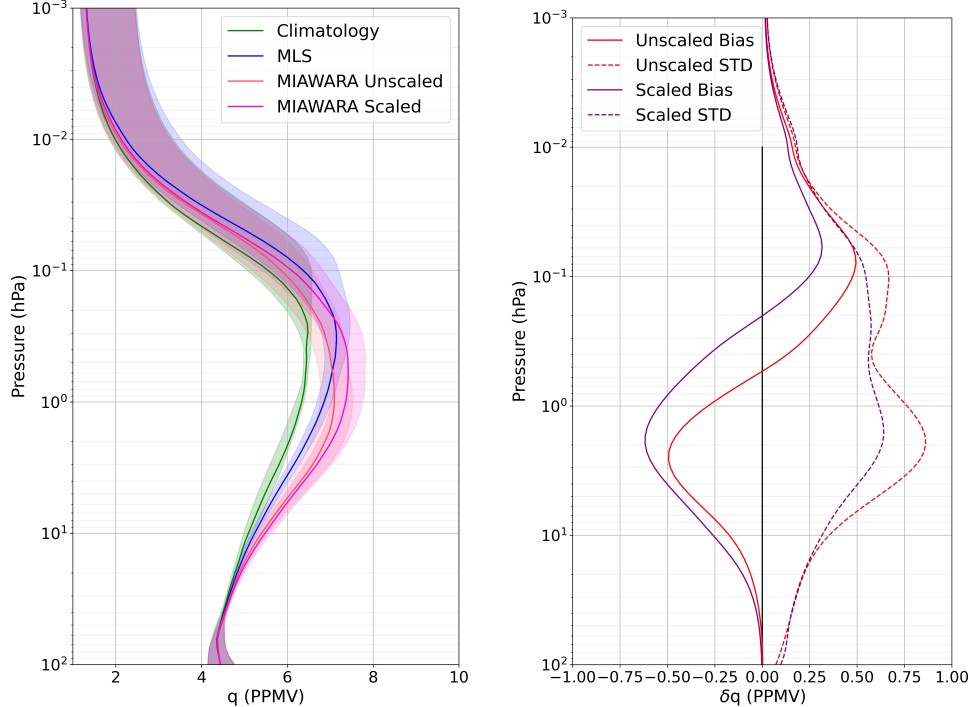

**Figure 10.** Median water vapour profiles from measurements provided by MLS, the uncorrected AC-240 spectra and the corrected AC-240 spectra (left). The 25th to 75th percentiles are shown in translucent shading of the corresponding colour; and the bias and standard deviation (MLS - MIAWARA) of the difference between the MLS measurement and the uncorrected AC-240, and corrected AC-240 measurements (right).

Figure 10 shows that the MIWARA estimates a higher water vapour mixing ratio than the MLS between $10\,\mathrm{hPa}$ and $0.2\,\mathrm{hPa}$,
and that the profile maxima occurs as slightly lower in altitude on the MIAWARA than the MLS. Although the bias in the retrieved water vapour with the corrected spectra is slightly degraded below $0.3\,\mathrm{hPa}$ relative to MLS, the inverse is true above this pressure level. The standard deviation of differences is also improved relative to the uncorrected differences, indicating that changes in water vapour are more consistent between MLS and the corrected measurements, compared to the uncorrected measurements.

**4 Conclusions**

An overview of a key upgrade to the MIAWARA instrument, the installation of a new spectrometer, has been presented, alongside analysis of the calibrated spectra and the effect on retrievals with this new spectrometer. A key finding was the need for a scaling correction of $4.6\,\%$ to the calibrated spectra of the AC-240 spectrometer which has been recording measurements on the MIAWARA instrument for the past 13 years and more.



The dataset presented is of use both in detecting water vapour trends in the middle atmosphere and provides another valuable reference to other instruments capable of measuring water vapour in the middle atmosphere, such as the MLS and the ACE-FTS.

The result of a scaling factor of $4.6\,\%$ was notably smaller than the scaling factor needed to correct the spectra in the work of Sauvageat et al. (2021), who found that an $8\,\%$ scaling factor was able to correct for the effects of spectral leakage in their 330 experiment. It is also less than the value of $7\,\%$ documented in Nedoluha et al. (2022). For the analysis presented here, the $4.6\,\%$ scaling factor seems to both bring the retrieved water vapour profile closer to the profiles retrieved from the USRP measurements, and improve the retrievals with respect to the Aura MLS measurements. Both these results indicate that this correction improves the accuracy of MIAWARA water vapour measurements.

As highlighted briefly in the earlier comparison of the USRP and AC-240 spectrometers, the years 2022 and 2023 have been 335 quite an exception in terms of the water vapour anomaly. This has arisen in the aftermath of the 2022 Hunga Tonga-Hunga Ha'apai volcano which emitted a plume posited to have reached up to the mesosphere (Proud et al., 2022). The observations from the MIAWARA, alongside other datasets, will be used to analyse the impact of the volcano on middle atmospheric water vapour and the possible impacts this has had. The MIAWARA dataset will also be used for the calculation of trends and monitoring of middle atmosphere water vapour.

The instrumental upgrade presented, combined with the unique environmental conditions following the 2022 Hunga Tonga-Hunga Ha'apai volcanic eruption, presents an opportunity to make a deeper understanding of atmospheric dynamics and the influences of major geophysical events on water vapor distributions in the middle atmosphere.

*Code availability.* The analysis codes supporting the findings of this study are openly available in the Git repository. These can be accessed at the following URL: https://github.com/alistairbbell/miawara_developments.

*Data availability.* Water vapour retrieval data is available from the NDACC database (https://ndacc.larc.nasa.gov/). All other data is available on request.

*Author contributions.* AB performed the code development and data analysis, and wrote the majority of this publication. Codes from ES were used in the calibration framework, and support was given on spectrometer comparisons. GS and KH provided support with the manuscript content. AM has provided supervision of the work.

*Competing interests.* The authors declare that they have no competing interests.



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
