# Peer review of "Developments on a 22GHz Microwave Radiometer and Reprocessing of 13-Year Time Series for Water Vapour Studies"

_EGUsphere, 2024_

## Referee Comment (RC1)

**Review of "Developments on a 22GHz Microwave Radiometer and Reprocessing of 13-Year Time Series for Water Vapour Studies" by Bell et al**

**1  General Comments**

Measurement of water vapour in the middle atmosphere is an important and difficult process; its long-term time series is valuable for climate studies. This paper provides 13-year ground-based measurements with attention to the calibration and comparison with the MLS satellite measurements. It should be published, subject to some minor corrections.

**2  Specific comments**

- Figure 2: What are the orange lines and grey shades for? It would be helpful to provide the information in the caption. And, "(right)" in the caption is confusing. There is only one plot for the MIAWARA time series.

- Figure 8 is shown but not discussed.

- Figure 9: Why is the summer bias larger? Why is the bias below/above 1 hPa negative/positive? The reason should be discussed.

- Figure 10: Putting the profiles before the bias in Figure 9 seems better for the readers to understand the bias pattern. But again, why is the peak altitude in MIAWARA lower than the MLS?

**3  Technical corrections**

- Line 128 and elsewhere: "H2O" should be "$H_2O$".

- Line 139 and elsewhere: "PPMv" or "PPMV" should be "ppmv".

- Line 194: There are two "that" in this sentence. The second one seems to be a grammar error.

- Line 285: See also papers by Millán et al. (2022), and Zhou et al. (2024).

  L. Millán, M. L. Santee, A. Lambert, N. J. Livesey, F. Werne, M. J. Schwartz, et al. (2022). The Hunga Tonga-Hunga Ha'apai Hydration of the Stratosphere. Geophysical Research Letters, 49, e2022GL099381, doi:10.1029/2022GL099381, 2022

  Zhou, X., Dhomse, S. S., Feng, W., Mann, G., Heddell, S., Pumphrey, H., et al. (2024). Antarctic vortex dehydration in 2023 as a substantial removal pathway for Hunga Tonga-Hunga Ha'apai water vapor. Geophysical Research Letters, 51, e2023GL107630. https://doi.org/10.102

---

## Author Comment (AC1)

Thanks to both reviewers for taking time to assess the paper and give some valueable feedback. Below are the original comments from the reviewers, with a response from the authors in green text and what has been changed in the article in response to the comment shown in red.

Comment: Figure 2: What are the orange lines and grey shades for? It would be helpful to provide the information in the caption. And, "(right)" in the caption is confusing. There is only one plot for the MIAWARA time series.

Added: The area between the two orange lines indicates where the measurements have a measurement response of above 0.6, meaning that most of the information comes from the atmospheric signal and not the a priori. The grey areas show where there were no measurements made of acceptable quality for more than five consecutive days.

Figure 8 is shown but not discussed.

Added: In figure \ref{fig:hist_difference_q}, the distribution of the differences between the unscaled AC-240 retrievals and the USRP retrievals, and the scaled AC-240 retrievals and the USRP retrievals, all at four pressure levels between 0.1 hPa and 1 hPa. The distribution of the differences of the corrected (scaled) and uncorrected (non-scaled) retrievals compared to those made with the USRP both exhibits roughly normal distributions. Whilst the standard deviation of differences are very similar for the two distributions at each pressure level, the scaled measurements have a median difference of closer to zero at every pressure level except \SI{1.1}{hPa}.

Figure 9: Why is the summer bias larger? Why is the bias below/above 1 hPa negative/positive? The reason should be discussed.

Response: The reason for the inverting bias cannot be known for sure; it could come from systematic errors in MLS measurements, MIAWARA measurements, or both. It is interesting that in the paper by Nedoluha et al. (2020), a similar pattern is found when comparing the WVMS to MLS and HALOE (seen in figure 8. [although this is WWMS - MLS whilst we show MLS - MIAWARA, so the sign of the bias is opposite.]). This work shows an overestimation of water vapour by the WWMS compared to MLS below 65km, and an underestimation over this height. The bias could therefore be a due to the difference in measurement technique (for example, common assumptions made in the retrieval of water vapour for ground-based radiometers compared to occultation/limb sounding), or biases in common components used in both radiometers, as could have been the case with the upgraded spectrometer in the MIAWARA.

With regards to the seasonal biases, this was also found to be the case with the SOMORA radiometer in Sauvageat et al. (2022). The seasonal bias is hypothesised as potentially due to a seasonal change in the instrumental baseline which had not been taken into account or increases in the optical depth (opacity) of the atmosphere in summer. Both points could be valid for the MIAWARA, but the most likely is the second point. In the calibration of measurements, a correction is made of the tropospheric attenuation of the measurements from a calculation of the tropospheric optical depth, which is in turn calculated from the tipping curve calibration of the cold sky view. An overestimation of the tropospherical optical depth could result in the over-correction of the tropospheric attenuation, which would lead to larger water vapour mixing ratios being retrieved from the measurement. As in summer, the optical depth of the troposphere around 22 GHz is larger than in the winter, any overestimation would have a bigger impact on the measurements in the summer months.

Added: A similar effect is seen in the ozone radiometer SOMORA \citep{sauvageat2022harmonized}, and has been hypothesised to come from a seasonal-dependent baseline, or due to changes in the optical depth of the troposphere in summer compared to winter. As the tropospheric optical depth is also used to correct for attenuation of middle atmosphere water vapour absorption line signal (see section \ref{section:calibration}), it is possible that errors in the calculation of the optical depth which increase in summer could result in the pattern seen in figure \ref{fig:MIA_MLS}.

Figure 10: Putting the profiles before the bias in Figure 9 seems better for the readers to understand the bias pattern. But again, why is the peak altitude in MIAWARA lower than the MLS?

Response: This is good point- that the altitude of the peak of the middle atmosphere water vapour pressure should be better understood. We hope with the new generation of ground-based radiometers, satellite instruments, and in-situ sensors that errors on each respective instrument will be better quantifiable. Due to the width of the averaging kernels at the upper heights, there is some contamination of the peak altitudes from areas with lower measurement response that seems to artificially reduce the MIAWARA water vapor peak altitude.

Change: order of figures changed as suggested

Line 128 and elsewhere: "H2O" should be "$H_2O$"

Change: as suggested

Line 139 and elsewhere: "PPMv" or "PPMV" should be "ppmv".

Change: as suggested

Line 194: There are two "that" in this sentence. The second one seems to be a grammar error.

Change: as suggested

**Bibliography**

Nedoluha, G., Maillard Barras, E., Haefele, A., Hocke, K., Kämpfer, N. and Boyd, I., 2020. Study of the dependence of long-term stratospheric ozone trends on local solar time. *Atmospheric Chemistry and Physics*, 20(12), pp.8453-8471

Sauvageat, E., Maillard Barras, E., Hocke, K., Haefele, A. and Murk, A., 2022. Harmonized retrieval of middle atmospheric ozone from two microwave radiometers in Switzerland. *Atmospheric Measurement Techniques*, 15(18), pp.6395–6417

---

## Author Comment (AC2)

Thanks to both reviewers for taking time to assess the paper and give some valueable feedback. Below are the original comments from the reviewers, with a response from the authors in green text and what has been changed in the article in response to the comment shown in red.

Comment: Line 18 – The references given here are fine, but generally this sentence would mention balloon borne measurements. They are mentioned later, but only for the lower stratosphere (FPH balloons). The next paragraph starts out with "Most obviously, radiosonde balloons measure in the troposphere". Perhaps the authors are specifically referring to remote measurements here, but they do not say this. A general reorganization of these introductory paragraphs would be helpful.

Response: Agreed that this section had some repetition and was not very readable. It has now been restructured as below

Change: There are various methods for measuring water vapour in the atmosphere. In the troposphere, routine observations are often made using radiosondes like the RS-41 \citep{Vaisala2022}. In the lower stratosphere, semi-routine measurements can be taken using in-situ balloon-borne devices \citep{hurst2011stratospheric,Graf_2021_ALBATROSS,Brunamonti_2023_ALBATROSS}, although continuous monitoring in this region typically depends on remote sensing technologies \citep{Sica_2016_RALMO_water_vapor}. In contrast, measuring water vapour in the upper stratosphere and mesosphere is more challenging due to the remoteness, extreme conditions, and the relatively low mass of water vapour in these layers. Due to these difficulties, microwave radiometry has become a common method for ground-based and satellite-borne instruments in these upper atmospheric regions \citep{Haefele_2008_water_vapor,Straub_2012_water_vapour_tracer,Schranz_2019}.

The Microwave Limb Sounder aboard the Aura satellite, launched in 2004, remains a key source of global water vapor data for the middle atmosphere \citep{waters2006earth}. However, the aging of this and other long-operating instruments has led to measurement drifts, making it difficult to distinguish between real atmospheric changes and artifacts from the instruments themselves \citep{livesey2021investigation}.

Line 39 – This statement is too strong. Even with just 2 measurements per day (one during the day and one at night), MLS is able to provide some information regarding diurnal variations.

Change: and there is limited possibility of capturing diurnal variations due to the spacing in local time between the two MLS overpasses

Line 53 – This sentence regarding pressure broadening is very awkward. The point is that the effect of pressure broadening is to increase the spectral width of the emission with increasing pressure.

Change: The effect of pressure broadening \citep{liebe1985updated} effectively increases the spectral width of emission with increasing atmospheric pressure. This allows the recorded spectral radiances can be inverted to retrieve vertical profiles of water vapour.

Line 69 – What is being described here is a Dicke switching scheme. The use of the word "calibration" here is very confusing.

Response: Agreed that when this is first explained that the quantity being explained (line view minus reference view) is indeed the same as Dicke switching. The paragraph has been re-written and the balancing calibration term introduced when the relation of the above quantity to hot and cold views are explained.

Change: In order to optimise the noise and linearity of the spectra, most middle atmosphere water vapour radiometers use a scheme where the antenna view rapidly switches in quick succession between the sky (line view) and a reference (reference view) \citep{forkman200322}.

The phrase "calibrated spectra" is used repeatedly. In almost all cases, replacing this phrase with simply "spectra" would reduce confusion.

Change: Deleted where not applicable

Line 180 – "This improvement stems from the 14-bit analog-to-digital converter (ADC), a notable step up from the AC240's 8-bit ADC, as well as advancements in digital signal processing that reduce numerical errors." – No evidence is presented that any of the difference shown are caused by the 8-bit nature of the ADC.

Response: Agreed that no evidence is presented to show that this is the cause of the improvement. However, it would make sense that a higher-resolution ADC reduces quantisation error, which means that the digital output more accurately reflects the analogue input across the entire range. The better precision could also reduce rounding errors when compared to the lower precision spectrometer.

Change: This improvement **could** stem from the 14-bit analog-to-digital converter (ADC)
Comment: Line 222 – With regards to problems with the central channels on the AC240 a reference to Gomez et al. 2012 (RS1010, doi:10.1029/2011RS004778) would be appropriate here.

Change: As suggested

Comment: Line 223- "This was potentially due to the relatively large noise levels in these calibrations due to the short integration times of several hours, compared to the operational integration time of one day." Perhaps I am wrong, however I think that this has nothing to do with the integration times, but with the fact that on the scale shown on Figure 3 one cannot detect ~5-10% differences in the ozone line.

Response: I would say actually that both are true - in this figure it would not be possible to see the 5-10% difference, but when the difference between the spectrometers is plotted, the noise (related to the integration time) means that this difference is also not visible.

Figure 5 – Since these 2 panels are being compared, please use the same ranges and ticks for the x-axes.

Change: Figure remade as suggested

To what extent is the difference in the high-altitude sensitivity in this figure is caused by the fact that the central 2 channels in the AC-240 are not being used? Or have I misunderstood something here?

Response: Yes, the fact that the central two channels are not used, and the fact that the USRP frequency resolution is five times that of the AC-240, essentially meaning that there are ten measurements of the line centre in the USRP that the AC-240 does not have.

The bump in the AVK's near 0.01 hPa in Figure 5 is very strange. It shows up in the AC-240 plot as

well, albeit not as clearly because retrievals from that spectrometer are not very sensitive near that pressure.  Is there perhaps a change in the thickness of the retrieved layers at this level?  Absent this, or some other a physically plausible explanation, it is difficult to believe that this is not indicative of an error in the retrieval code.

This is due to the shape of the a priori errors which are specified in the retrieval. On Figure 3, you can see that around 0.1hPa, the is a large oscillation in the standard deviation in MLS measurements at this height. The errors prescribed to the algorithm were trialed using only the smoothed (green) curve initially, and this led to even larger oscillations in the averaging kernels.

Figure 9 – A single contour plot here would probably be sufficient.  The pattern is the same for both spectra.

Change: Deleted first plot as suggested

Figures 10 – Given that the only difference in these two comparisons is a scaling of the bias, it is very surprising that there is a difference in the scaled and unscaled STD at this level in this Figure. Presumably this occurs because there are differences in the profiles or spectra being used in the comparison.  If this is the case please state this.   If this is not the case then please provide another explanation.

Response: The figures show statistics of retrievals made after scaling the spectra with the 4.6% correction factor. As can be seen on figure 8, in which scaled and non-scaled retrievals are compared to USRP retrievals, the effect of the scaling does not simply shift the distribution of retrieved water vapour, although in this figure, the standard deviation of errors do not seem to change by much.

Comment: Line 336 – The mesospheric H2O observed in these measurements is unrelated to the direct injection into the mesosphere noted in Proud et al. (2022).  I recommend a reference to Nedoluha et al. (2024)  https://doi.org/10.1029/2024JD040907

Change: As suggested

**Bibliography**

Nedoluha, G., Maillard Barras, E., Haefele, A., Hocke, K., Kämpfer, N. and Boyd, I., 2020. Study of the dependence of long-term stratospheric ozone trends on local solar time. *Atmospheric Chemistry and Physics*, 20(12), pp.8453-8471

Sauvageat, E., Maillard Barras, E., Hocke, K., Haefele, A. and Murk, A., 2022. Harmonized retrieval of middle atmospheric ozone from two microwave radiometers in Switzerland. *Atmospheric Measurement Techniques*, 15(18), pp.6395–6417